# Intensive care unit patients' opinion on enrollment in clinical research: A multicenter survey

**Flavia Julie do Amaral Pfeilsticker[1], Carolina Aguiar Sant Anna Siqueri[1], Niklas Soderberg Campos[1], Fernanda Guimarães Aguiar[1], Maria Laura Romagnoli[1], Renato Carneiro de Freitas Chaves[1], Carolina Scoqui Guimarães[2], Adriano José Pereira[1,2], Ricardo Luiz Cordioli[1], Ary Serpa Neto[1], Murillo Santucci Cesar Assunção[1], Thiago Domingos Corrêa[1] ***

**1** Department of Critical Care Medicine, Hospital Israelita Albert Einstein, São Paulo, Brazil, **2** Intensive Care Unit, Hospital Municipal Vila Santa Catarina, São Paulo, Brazil

* thiago.correa@einstein.br

**Data Availability Statement:** The study data bank has been deposited at Dryard (https://doi.org/10.5061/dryad.66t1g1k06).

## Abstract

### Background

In most emergency situations or severe illness, patients are unable to consent for clinical trial enrollment. In such circumstances, the decision about whether to participate in a scientific study or not is made by a legally designated representative.

### Objective

To address the willingness of patients admitted to the intensive care unit (ICU) to be enrolled in a scientific study as volunteers, and to assess the agreement between patients' and their legal representatives' opinion concerning enrollment in a scientific study.

### Methods

This survey was conducted in two hospitals in São Paulo, Brazil. Patients (≥18 years) with preserved cognitive functions accompanied by a surrogate admitted to the ICU were eligible for this study. A survey containing 28 questions for patients and 8 questions for surrogates was applied within the first 48h from ICU admission. The survey for patients comprised three sections: demographic characteristics, opinion about participation in clinical research and knowledge about the importance of research. The survey for legal representatives contained two sections: demographic characteristics and assessment of legal representatives' opinion in authorizing patients to be enrolled in research.

### Results

Between January 2017 and May 2018, 208 pairs of ICU patients and their respective legal representatives answered the survey. Out of 208 ICU patients answering the survey, 73.6% (153/208) were willing to be enrolled in the study as volunteers. Of those patients, 65.1% (97/149) would continue participating in a research even if their legal representative did not support their enrollment. Agreement between patients' and surrogates' opinion concerning

**Funding:** The author(s) received no specific funding for this work.

**Competing interests:** he authors have declared that no competing interests exist.

participation was poor [Kappa = 0.11 (IC95% -0.02 to 0.25)]. If a consent for study participation had been obtained, 69.1% (103/149) of patients would continue participating in the study until its conclusion, and 23.5% (35/149) would allow researchers to use data collected to date, but would withdraw from the study on that occasion.

## Conclusion

The majority of patients admitted to the ICU were willing to be enrolled in a scientific study as volunteers, also after a deferred informed consent procedure has been used. Nevertheless, contradictory opinions between patients and their and their legal representatives' concerning enrollment in a scientific study were often observed.

## Introduction

Scientific research involving human beings has been the cornerstone of the development of medical knowledge. Nevertheless, in order to preserve the principle of autonomy of research subjects as stated in Helsinki Declaration [1], study investigators must guarantee that an informed consent is obtained from each study participant before enrollment [2]. However, life-threatening conditions, commonly observed among patients admitted to the intensive care unit (ICU), may preclude ICU patients from consenting enrollment in a clinical trial [3, 4]. In such circumstances, consent must be obtained from a legally authorized representative [1].

The acquisition of consent by proxies can be challenging in the critical care setting. In many circumstances during the ICU stay, legally authorized representatives may not be readily available or may not even exist, precluding conduction of a clinical trial in emergency situations with a narrow therapeutic window [2]. Thus, depending on international and local ethics committees' policies, consent can be waived or deferred by patient (i.e., patient deferred consent) or by proxy (i.e., proxy deferred consent) [3]. When a deferred consent is obtained, the patient is enrolled in the study but the informed consent must be obtained as soon as possible by patient or his/her legal representative [3].

The discussion regarding the use of deferred proxy consent instead of regular consent is reasonable, considering that surrogates' decision may not reflect patients' opinion [5]. Moreover, critical care conditions may trigger emotional stress disorders [6], religious concerns [7] and financial problems [8], which commonly affect relatives' decision making. Despite the relevance of this topic, Brazil is lacking in studies addressing willingness of patients admitted to the ICU to be enrolled in a scientific study as volunteers. Moreover, agreements between ICU patients' and their legal representatives' opinion concerning enrollment in a scientific study is unknown.

## Objectives

Our purpose was to address the willingness of patients admitted to the ICU to be enrolled in a scientific study as volunteers, and to assess the agreement between ICU patients' and their legal representatives' opinion concerning enrollment in a scientific study.

## Material and methods

### Study design and settings

This prospective observational study was conducted in two ICUs located in two hospitals (private and public) in São Paulo, Brazil. The Ethics Committee of Hospital Israelita Albert

Einstein (CAAE: 62100416.8.1001.0071) and of Secretaria Municipal da Saúde de São Paulo–SMS/SP (Municipal Secretary of Health of São Paulo) (CAAE: 62100416.8.3001.0086) approved the study protocol. Written informed consent was obtained from each study participant. This study was reported in accordance with The Strengthening the Reporting of Observational Studies in Epidemiology (STROBE) guidelines [9].

## Characteristics of study participants

Adult patients (≥18 years old) with preserved cognitive function accompanied by a surrogate admitted to the ICU were eligible for this study. Patients under 18 years old, patients unable to understand or to speak Portuguese, patients with no legal representative during ICU stay, patients with impaired consciousness at the time of evaluation (i.e., sedated patients or patients with delirium), and patients with developmental delay were excluded.

## The survey

The questionnaire was developed by the authors (FJAP, RLC and TDC). Content validation was performed by a panel of senior researchers (AJP, ASN and MSCA). They critically revised the survey instrument and judged whether the instrument meets its goal. The panel of senior researchers sent the survey for the developers to make the necessary adjustments. These adjustments were performed and the revised survey was reassessed by the panel. After the third revision, content validity was supported by the panel. Face validity, i.e, whether the survey instrument appears to test what it is supposed to, and questionnaire's psychometric properties were tested by all other co-authors and by a small group of volunteers (10 employers) in both ICUs.

The survey for patients comprised 28 questions distributed in three sections—demographic characteristics, opinion about participation in clinical research and knowledge of the importance of research, respectively (Supporting Information). The survey for legal representatives contained 8 questions distributed in two sections—demographic characteristics and assessment of the legal representatives' opinion on authorizing patients to be enrolled in research (Additional file 1).

Participants who accepted to participate in this study responded to a traditional paper and pencil questionnaire, which was then put into an opaque envelope to preserve their anonymity. The survey was simultaneously applied to patients and their legal representative within the first 48 hours of ICU admission. Once responded, the answers were not able to be revised. The study database was structured in the Research Electronic Data Capture (REDCap) [10], hosted in a private and safe server at Hospital Israelita Albert Einstein, which was filled out by the investigators. No financial incentives were offered for participants of this study. A threshold of more than 90% of answers was used to determine completion of questionnaires.

## Data analysis

To achieve 95% confidence and 5% precision by adopting a conservative sample (50% of the participants answering YES to the question "Would you participate in a clinical trial as a volunteer?"), an estimated sample size of 208 pairs of patients and legal representatives was determined.

Participants were pooled into two groups according to their willingness to participate as volunteers in a scientific study (Yes/Probably yes) and (No/Probably no) and accordingly to the type of hospital (Private vs. Public). Categorical variables were displayed as absolute and relative frequencies. Numerical variables were presented as mean and standard deviation (SD)

or median with interquartile range (IQR) in case of non-normal distribution, tested with Kolmogorov-Smirnov test.

Comparisons were performed between the pooled groups (Yes/Probably yes) and (No/Probably no) and accordingly to the type of hospital (Private vs. Public). Categorical variables were compared with chi-square test. Continuous variables were compared using independent t test or Mann-Whitney U test in case of non-normal distribution, tested by the Kolmogorov-Smirnov test. Agreement between patients and their legal representatives was assessed using kappa statistics. Kappa value closer to zero should be interpreted as no agreement, whereas Kappa value close to one should be interpreted as perfect agreement.

A multivariable logistic regression analysis was performed to address patients' characteristics associated with their willingness to participate as volunteers in a scientific study. Variables considered for the multivariate modeling included age, gender, educational level, religion and family income. Results were presented as odds ratio (OR) with 95% confidence interval (95% CI). The performance of the model was evaluated by assessing discrimination and calibration. Discrimination was evaluated with the area under the receiver operating characteristics curve (AUROC) and calibration was evaluated with the Hosmer-Lemeshow test.

Two-tailed tests were used and when p<0.05, the test was considered statistically significant. No adjustment was made for missing data. The SPSS™ (IBM™ Statistical Package for the Social Science version 23.0) was used for statistical analyses and GraphPad Prism version 7.0 (GraphPad Software, California, USA) was used for graph plotting.

## Results

### Participants' characteristics

Between January 2017 and May 2018, 208 pairs of ICU patients and their respective legal representatives answered the survey [87.0% (181/208) private hospital and 13.0% (27/208) public hospital]. The median (IQR) age of patients was 60 (43–75) years, 49% were female and 56.7% had a high educational level. The median (IQR) age of legal representatives was 60 (43–75) years, 65.9% were female and 68.8% were next of kin (spouse/son/daughter). The characteristics of study participants according to their willingness to participate as volunteers in a scientific study and according to the type of hospital (private vs. public) are presented, respectively, in Table 1 and S1 Table in S1 File.

### Patients' opinion about participation in clinical research

Out of 208 ICU patients who answered the survey, 73.6% (153/208) were willing to be enrolled in a scientific study as volunteers (Fig 1 and S2 Table in S1 File). Approximately 14.0% of patients [13.9% (29/208)] who answered the survey had never been enrolled in a scientific study. The willingness to be enrolled in a scientific study as volunteers did not differ between patients that had already been enrolled in a scientific study as a volunteer compared to those patients that had never been enrolled in a scientific study [86.2% (25/29) vs. 71.5% (128/179), respectively, p = 0.096].

Less than one third of the patients [28.0% (42/150)] that were willing to be enrolled in a scientific study as a volunteer had ever heard of informed consent. The most common reason given by patients for choosing to be volunteers in a scientific study was the fact they believed the study results would bring benefits to the general population in the future [88.2% (135/153) of patients] (Table 2).

Approximately half of the patients [48.3% (72/149 patients) would not change their opinion, i.e., no longer accept to participate in a scientific study if their legal representative did not

**Table 1. Characteristics of study participants according to their willingness to participate as volunteers in a scientific study.** Data presented as median (interquartile range) or nº/total (%)[#].

| Characteristics | All patients (N = 208) | Yes / Probably yes (N = 153) | No / Probably no (N = 55) | P value |
|---|---|---|---|---|
| Age, years | 60 (43–75) | 58 (43–70) | 70 (46–82) | 0.002[a] |
| Female, gender | 102/208 (49.0) | 69/153 (45.1) | 33/55 (60.0) | 0.058[b] |
| Educational level | | | | 0.803[b] |
| Master's / PhD | 21/208 (10.1) | 17/153 (11.1) | 4/55 (7.3) | |
| Higher education | 118/208 (56.7) | 85/153 (55.6) | 33/55 (60.0) | |
| High school | 40/208 (19.2) | 30/153 (19.6) | 10/55 (18.2) | |
| Primary school | 27/208 (13.0) | 19/153 (12.4) | 8/55 (14.5) | |
| Illiterate | 2/208 (1.0) | 2/153 (1.3) | 0/55 (0.0) | |
| Religion | | | | 0.013[b] |
| Catholic | 145/208 (69.7) | 106/153 (69.3) | 39/55 (70.9) | |
| Other | 20/208 (9.6) | 15/153 (9.8) | 5/55 (9.1) | |
| Evangelic | 15/208 (7.2) | 13/153 (8.5) | 2/55 (3.6) | |
| No religion | 15/208 (7.2) | 14/153 (9.2) | 1/55 (1.8) | |
| Jewish | 13/208 (6.3) | 5/153 (3.3) | 8/55 (14.5) | |
| Place of residence | | | | 0.972[b] |
| Southeast | 173/208 (83.2) | 127/153 (83.0) | 46/55 (83.6) | |
| South | 16/208 (7.7) | 11/153 (7.2) | 5/55 (9.1) | |
| Central-west | 10/208 (4.8) | 8/153 (5.2) | 2/55 (3.6) | |
| Northeast | 5/208 (2.4) | 4/153 (2.6) | 1/55 (1.8) | |
| North | 4/208 (1.9) | 3/153 (2.0) | 1/55 (1.8) | |
| Family income | | | | 0.708[b] |
| >10 minimum wages | 133/198 (67.2) | 97/146 (66.4) | 36/52 (69.2) | |
| 6–10 minimum wages | 24/198 (12.1) | 16/146 (11.0) | 8/52 (15.4) | |
| 2–5 minimum wages | 33/198 (16.7) | 27/146 (18.5) | 6/52 (11.5) | |
| 1 minimum wage | 7/198 (3.5) | 5/146 (3.4) | 2/52 (3.8) | |
| No income | 1/198 (0.5) | 1/146 (0.7) | 0/52 (0.0) | |
| **Legal Representatives** | | | | |
| Age, years | 49 (37–60) | 48 (37–61) | 50 (38–58) | 0.781[a] |
| Female, gender | 137 (65.9) | 104/153 (68.0) | 33/55 (60.0) | 0.285[b] |
| Kinship degree | | | | 0.328[b] |
| Husband / wife | 75 (36.1) | 57/153 (37.3) | 18/55 (32.7) | |
| Son / daughter | 68 (32.7) | 48/153 (31.4) | 20/55 (36.4) | |
| Parent | 25 (12.0) | 21/153 (13.7) | 4/55 (7.3) | |
| Brother / sister | 15 (7.2) | 12/153 (7.8) | 3/55 (5.5) | |
| Other | 25 (12.0) | 15/153 (9.8) | 10/55 (18.2) | |

#: For variables with missing data, summary data are based on available cases. P values were calculated with (a): Mann-Whitney U test or (b) chi-square test.

support their enrollment, while 50% (75/150 patients) would change their opinion, i.e., no longer accept to participate in a scientific study if their primary attending physician did not support their enrollment (Table 2). Patients are more likely to accept to participate as volunteers in a scientific study only when it involves data collection from their medical records [92.7% (140/151)] or when a new drug is being tested [63.6% (96/151)] rather than when a new surgical treatment is being studied [58.6% (88/150)] (Table 2).

Finally, most of the patients [90.7% (136/150)] who would accept to be enrolled in a scientific study as volunteers would like to be informed about the final result of the scientific study

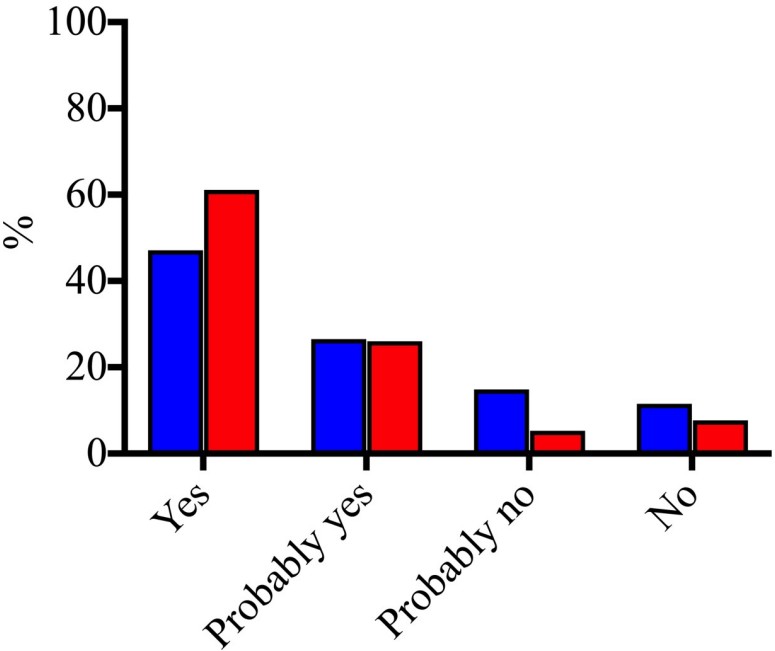

**Fig 1. Patients' and legal representatives' answers to the respective questions "Would you like to be enrolled in a scientific study as a volunteer?" and "Would you authorize your relative to participate as a volunteer in scientific research during their stay in the ICU?".** Blue bars represent patients and red bars represent legal representatives.

they had been enrolled in. The survey responses provided by patients that would not participate as volunteers in a scientific study are presented in S3 Table in S1 File.

## Patients' knowledge about the importance of research

Ninety-two percent (185/201) of patients agreed that in order to develop new treatments, research involving human beings must be conducted (S4 Table in S1 File). While 92.8% of patients trust results obtained by research conducted in private hospitals, only 66.5% and 54.1%, respectively, believed in results obtained by public hospitals and by the pharmaceutical industry (S4 Table in S1 File).

## Patients' characteristics and their willingness to participate as volunteers

In a multivariable logistic regression analysis, only patients age ≥65 years (OR, 0.34; 95%CI, 0.14 to 0.67; p = 0.002) were independently associated with a lower odds of a patient to participate as a volunteer in scientific research (Table 3).

## Patients' opinion about deferred consent

Out of 153 ICU patients willing to be enrolled in a scientific study as volunteers (Table 2), 70.2% (106/151) of patients would trust a relative with the decision to be enrolled in a scientific study (Table 2). The agreement between patients' and surrogates' opinion concerning participation as volunteers in a scientific study was poor [Kappa = 0.11 (IC95% -0.02 to 0.25); p = 0.071)] (S2 Table in S1 File).

If a consent for study participation had been obtained, 69.1% (103/149) of patients would continue participating in the study until its conclusion, 23.5% (35/149) would allow researchers to use data collected to date, but would withdraw from the study on that occasion, and

**Table 2. Patients willing (Yes / Probably yes) to be enrolled in a scientific study opinion about participation in clinical research accordingly to the type of hospital.** Data presented n°/total (%)#.

| | All patients (N = 153) | Private Hospital (N = 130) | Public Hospital (N = 23) | P value |
|---|---|---|---|---|
| Have you ever been enrolled in a scientific study? | | | | 0.021 |
| Yes | 25/153 (16.3) | 25/130 (19.2) | 0/23 (0.0) | |
| No | 128/153 (83.7) | 105/130 (80.8) | 23/23 (100.0) | |
| What motivates you to be a volunteer in a scientific study? | | | | |
| The study offers benefits for the general population in the future | 135/153 (88.2) | 114/130 (87.7) | 21/23 (91.3) | 0.743 |
| The study offers benefits to you immediately/in the future | 101/153 (66.0) | 88/130 (67.7) | 13/23 (56.5) | 0.342 |
| Your physician's request | 93/153 (60.8) | 79/130 (60.8) | 14/23 (60.9) | 1.000 |
| No treatment options available | 100/153 (65.3) | 89/130 (68.5) | 11/23 (47.8) | 0.062 |
| The study offers financial gains for your participation | 24/153 (15.7) | 19/130 (14.6) | 5/23 (21.7) | 0.533 |
| Would you change your opinion, i.e., no longer accept to participate in a scientific study if your legal representative did not support your enrollment? | | | | |
| Yes | 34/149 (22.8) | 29/127 (22.8) | 5/22 (22.7) | 0.670 |
| Probably yes | 18/149 (12.1) | 16/127 (12.6) | 2/22 (9.1) | |
| Probably no | 25/149 (16.8) | 23/127 (18.1) | 2/22 (9.1) | |
| No | 72/149 (48.3) | 59/127 (46.5) | 13/22 (59.1) | |
| Would you change your opinion, i.e., no longer accept to participate in a scientific study if your primary attending physician did not support your enrollment? | | | | 0.892 |
| Yes | 75/150 (50.0) | 65/128 (50.8) | 10/22 (45.5) | |
| Probably yes | 28/150 (18.7) | 23/128 (18.0) | 5/22 (22.7) | |
| Probably no | 16/150 (10.6) | 13/128 (10.2) | 3/22 (13.6) | |
| No | 31/150 (20.7) | 27/128 (21.1) | 4/22 (18.2) | |
| Would you participate as a volunteer in a scientific study involving medical data collection from your records or information on ICU routine and treatments? | | | | 0.727 |
| Yes | 126/151 (83.4) | 107/129 (82.9) | 19/22 (86.4) | |
| Probably yes | 14/151 (9.3) | 13/129 (10.1) | 1/22 (4.5) | |
| Probably no | 2/151 (1.3) | 2/129 (1.6) | 0/22 (0.0) | |
| No | 9/151 (6.0) | 7/129 (5.4) | 2/22 (9.1) | |
| Would you participate as a volunteer in a scientific study involving a new drug? | | | | 0.016 |
| Yes | 69/151 (45.7) | 53/129 (41.1) | 16/22 (72.7) | |
| Probably yes | 27/151 (17.9) | 25/129 (19.4) | 2/22 (9.1) | |
| Probably no | 27/151 (17.9) | 23/129 (17.8) | 4/22 (18.2) | |
| No | 28/151 (18.5) | 28/129 (21.7) | 0/22 (0.0) | |
| Would you participate as a volunteer in a scientific study of a new surgical treatment? | | | | 0.152 |
| Yes | 51/150 (34.0) | 40/128 (31.3) | 11/22 (50.0) | |
| Probably yes | 37/150 (24.6) | 32/128 (25.5) | 5/22 (22.7) | |
| Probably no | 31/150 (20.7) | 26/128 (20.3) | 5/22 (22.7) | |
| No | 31/150 (20.7) | 30/128 (23.4) | 1/22 (4.5) | |
| Would you trust a relative with the decision to be enrolled in a scientific study | | | | 0.727 |
| Yes | 88/151 (58.3) | 75/129 (58.1) | 13/22 (59.1) | |
| Probably yes | 18/151 (11.9) | 16/129 (12.4) | 2/22 (9.1) | |
| Probably no | 12/151 (7.9) | 9/129 (7.0) | 3/22 (13.6) | |
| No | 33/151 (21.9) | 29/129 (22.5) | 4/22 (18.2) | |

#: For variables with missing data, summary data are based on available cases. P values were calculated with chi-square test.

**Table 3. Multivariate logistic regression analysis addressing patients' characteristics associated with their willingness to participate as volunteers in a scientific study.**

| Characteristics | OR | 95% CI | P value |
|---|---|---|---|
| Age, years | | | |
| <65 | Reference | | |
| > = 65 | 0.34 | 0.17–0.67 | 0.002 |
| Gender | | | |
| Female | Reference | | |
| Male | 1.94 | 0.99–3.79 | 0.054 |
| Educational level | | | |
| Illiterate / primary school | Reference | | |
| High school | 1.01 | 0.30–3.44 | 0.991 |
| Higher education / Master's / PhD | 0.67 | 0.23–2.00 | 0.476 |
| Religion | | | |
| Others / None | Reference | | |
| Catholic | 1.00 | 0.49–2.05 | 0.988 |
| Family income | | | |
| < = 10 minimum wages | Reference | | |
| >10 minimum wages | 0.95 | 0.44–2.05 | 0.907 |

Participants were pooled into two groups according to their willingness to participate as volunteers in a scientific study (Yes/Probably yes) and (No/Probably no). OR: odds ratio, 95% CI: 95% confidence interval. The multivariate model had an area (95%CI) under the receiver operating characteristic curve of 0.66 (0.58–0.74) and a Hosmer-Lemeshow chi-square of 5.570 (p = 0.591).

7.4% (11/149) would withdraw from the study on that occasion and would not allow researchers to use data collected to date.

## Legal representatives' answers

Out of 208 legal representatives who answered the survey, 87% (181/208) of the participants would allow their relative to participate as volunteers in scientific research during their stay in the ICU (Fig 1 and S2 Table in S1 File).

Legal representatives were more prone to allowing their relative to participate as volunteers in scientific research when only data collected from their medical records was being tested than when a medical or surgical treatment was being tested (Table 4). Legal representatives at a public hospital were more prone to allowing their relatives to participate as volunteers in scientific research testing a new medical or surgical therapy compared to legal representatives at a private hospital (Table 4).

If a consent for study participation had been obtained and the respective relative did not regain consciousness, 63.3% (46/188) of legal representatives would allow their relative to continue participating in the study until its conclusion (Table 4).

## Discussion

In this multicenter survey conducted in two medical-surgical ICUs including 208 pairs of ICU patients and their respective legal representatives, we demonstrated that nearly three out of four ICU patients surveyed would like to be enrolled in a scientific study as volunteers. We also demonstrated that surrogates' opinion poorly reflects patients' opinion concerning enrollment in clinical research, since agreement between patients' and legal authorized representatives' opinion was poor.

**Table 4. Legal representatives' answers to survey accordingly to the type of hospital.** Data presented nº/total (%)[#].

| | All legal representatives (N = 208) | Private Hospital (N = 181) | Public Hospital (N = 27) | P value |
|---|---|---|---|---|
| Would you authorize your relative to participate as a volunteer in scientific research during their stay in the ICU? | | | | 0.510 |
| Yes | 127/208 (61.1) | 107/181 (59.1) | 20/27 (74.1) | |
| Probably yes | 54/208 (26.0) | 49/181 (27.1) | 5/27 (18.5) | |
| Probably no | 11/208 (5.3) | 10/181 (5.5) | 1/27 (3.7) | |
| No | 16/208 (7.7) | 15/181 (8.3) | 1/27 (3.7) | |
| Would you authorize your relative to participate as a volunteer in scientific research involving only data collection from medical records and/or data comprising ICU care and treatment routine? | | | | |
| Yes | 139/190 (73.2) | 116/164 (70.7) | 23/26 (88.5) | 0.282 |
| Probably yes | 39/190 (20.5) | 37/164 (22.6) | 2/26 (7.7) | |
| Probably no | 10/190 (5.3) | 9/164 (5.5) | 1/26 (3.8) | |
| No | 2/190 (1.1) | 2/164 (1.2) | 0/26 (0.0) | |
| Would you authorize your relative to participate as a volunteer in scientific research involving a new medication? | | | | 0.018 |
| Yes | 58/190 (30.5) | 46/164 (28.0) | 12/26 (46.2) | |
| Probably yes | 66/190 (34.7) | 54/164 (32.9) | 12/26 (46.2) | |
| Probably no | 47/190 (24.8) | 45/164 (27.4) | 2/26 (7.7) | |
| No | 19/190 (10.0) | 19/164 (11.6) | 0/26 (0.0) | |
| Would you authorize your relative to participate as a volunteer in scientific research involving a new surgical treatment? | | | | 0.031 |
| Yes | 46/188 (24.5) | 36/162 (22.2) | 10/26 (38.5) | |
| Probably yes | 72/188 (38.3) | 59/162 (36.4) | 13/26 (50.0) | |
| Probably no | 52/188 (27.7) | 50/162 (30.9) | 2/26 (7.7) | |
| No | 18/188 (9.6) | 17/162 (10.5) | 1/26 (3.8) | |
| In situations of emergency/urgency (loss of consciousness, cardiac arrest, etc) it is not possible to request authorization (consent) from patient or from legal representative about his/her participation in scientific research. If your relative did not regain consciousness and if you were informed that he/she was included in scientific research, would you? | | | | 0.452 |
| Allow him/her to continue participating until research was concluded | 119/188 (63.3) | 100/162 (61.7) | 19/26 (73.1) | |
| Allow only data collected to date to be used and would request his/her removal from research | 52/188 (27.7) | 46/162 (28.4) | 6/26 (23.1) | |
| Request his/her immediate removal from research and would not allow use of any data collected | 17/188 (9.0) | 16/162 (9.9) | 1/26 (3.8) | |

#: For variables with missing data, summary data are based on available cases.

The main purpose of informed consent in the context of clinical research is to ensure that ethical principles of autonomy and the self-determination of participating subjects are preserved [11, 12]. However, our findings support the hypothesis that informed consent obtained in emergency situations by surrogates may not fully reflect patients' willingness to participate in scientific research as volunteers [13–17].

Similar findings were reported by Newman and cols. in a survey conducted with sixty-nine adult patients and surrogates admitted to a medical ICU in the United States [14]. The authors demonstrated a marked discrepancy between patients' and surrogates' willingness to participate as volunteers in scientific research [14]. More interestingly, discrepancy between patients and surrogates opinion increased as complexity of hypothetical studies increased, ranging from less than 5% in an observational study involving only demographic and clinical data collection up to approximately 50% in a randomized controlled clinical trial [14].

In another study conducted in ten ICUs in France, two hypothetical research studies with different levels of risk to patients were applied to patients and their surrogates on the day of ICU discharge to the wards [13]–one involving minimal risk (a hypothetical study comparing crystalloids to colloids for volume expansion in septic shock), and another, greater-than-minimal risk to patients (a hypothetical study comparing early vs. late tracheotomy in patients requiring invasive mechanical ventilation). The authors reported that surrogates' opinion underestimated patients' wishes concerning their willingness to participate in scientific research, with a discrepancy rate of 32% and 42% between patients and surrogates, respectively, for minimal risk and greater-than-minimal risk hypothetical studies [13].

This study found that the most common reason for ICU patients to consent to participate as volunteers in a scientific study was the fact that they believed that in the future other patients might benefit from the study results. Similar findings were reported by Mehta and cols., who studied 96 surrogates' (substitute decision makers) motivations to provide consent or not for their critically ill adult family members to be enrolled in scientific research as volunteers [18]. The authors reported that the vast majority of substitute decision makers (91%) would agree with enrollment as they believe that the study results will help others in the future [18].

Our study has limitations. First, this survey was designed with predefined answers i.e., yes / probably yes / probably no / no, which might have compromised the judgement and perceptions of participants. Nevertheless, when compared to similar surveys in literature, it involves a remarkable number of responders [14] besides being the first survey carried out comparing the opinion of patients and their legal representatives in Latin America. Secondly, all surveys were applied by the study investigators; therefore, participants might have provided favorable answers, thus leading to bias. Thirdly, we excluded unconscious patients in this study. Thus, our results need to be interpreted with caution when considering more severe patients, whose legal representatives may have been experiencing severe emotional stress, which would compromise their decision making. Fourth, the survey was structured with a logical sequence of questions, which precluded bias minimization by randomizing or alternating the sequence of questions. Finally, most patients and their relatives were from a private hospital, which may limit the external validity of our results.

## Conclusion

The majority of patients admitted to the ICU were willing to be enrolled in a scientific study as volunteers, also after a deferred informed consent procedure has been used. Nevertheless, contradictory opinions between patients and their and their legal representatives' concerning enrollment in a scientific study were often observed. Efforts should be made in order to improve knowledge of ICU patients and their relatives concerning the importance and value of medical research to improve patients' engagement in clinical research.

## Supporting information

**S1 File.**
(DOCX)

## Acknowledgments

We thank Helena Spalic for proofreading this manuscript.

## Author Contributions

**Conceptualization:** Flavia Julie do Amaral Pfeilsticker, Renato Carneiro de Freitas Chaves, Adriano José Pereira, Thiago Domingos Corrêa.

**Data curation:** Flavia Julie do Amaral Pfeilsticker, Carolina Aguiar Sant Anna Siqueri, Niklas Soderberg Campos, Fernanda Guimarães Aguiar, Renato Carneiro de Freitas Chaves, Adriano José Pereira, Ricardo Luiz Cordioli, Thiago Domingos Corrêa.

**Formal analysis:** Flavia Julie do Amaral Pfeilsticker, Niklas Soderberg Campos, Renato Carneiro de Freitas Chaves, Ricardo Luiz Cordioli, Thiago Domingos Corrêa.

**Investigation:** Flavia Julie do Amaral Pfeilsticker, Carolina Aguiar Sant Anna Siqueri, Niklas Soderberg Campos, Fernanda Guimarães Aguiar, Maria Laura Romagnoli, Renato Carneiro de Freitas Chaves, Carolina Scoqui Guimarães, Adriano José Pereira, Ricardo Luiz Cordioli, Ary Serpa Neto, Murillo Santucci Cesar Assuncão, Thiago Domingos Corrêa.

**Methodology:** Flavia Julie do Amaral Pfeilsticker, Renato Carneiro de Freitas Chaves, Adriano José Pereira, Ary Serpa Neto, Murillo Santucci Cesar Assuncão, Thiago Domingos Corrêa.

**Project administration:** Thiago Domingos Corrêa.

**Supervision:** Thiago Domingos Corrêa.

**Writing – original draft:** Flavia Julie do Amaral Pfeilsticker, Thiago Domingos Corrêa.

**Writing – review & editing:** Flavia Julie do Amaral Pfeilsticker, Carolina Aguiar Sant Anna Siqueri, Niklas Soderberg Campos, Fernanda Guimarães Aguiar, Maria Laura Romagnoli, Renato Carneiro de Freitas Chaves, Carolina Scoqui Guimarães, Adriano José Pereira, Ricardo Luiz Cordioli, Ary Serpa Neto, Murillo Santucci Cesar Assuncão, Thiago Domingos Corrêa.

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
