## [Decision Letter · Decision Letter 0]

26 May 2020

PONE-D-20-10276

Intensive care unit patients’ opinion on enrollment in clinical research: a multicenter survey

PLOS ONE

Dear Dr. Corrêa,

Thank you for submitting your manuscript to PLOS ONE. After careful consideration, we feel that it has merit but does not fully meet PLOS ONE’s publication criteria as it currently stands. Therefore, we invite you to submit a revised version of the manuscript that addresses the points raised during the review process.

We look forward to receiving your revised manuscript.

Kind regards,

Andrew Carl Miller

Academic Editor

PLOS ONE

Journal Requirements:

"This prospective observational study was conducted in two ICUs located in two hospitals (private and public) in São Paulo, Brazil after obtaining institutional approval at each site (CAAE: 62100416.8.1001.0071). Written informed consent was obtained from each study participant."  

Once you have amended this statement in the Methods section of the manuscript, please add the same text to the “Ethics Statement” field of the submission form (via “Edit Submission”).

Additional Editor Comments (if provided):

Please see the reviewer comments.

Reviewers' comments:

Reviewer's Responses to Questions

**Comments to the Author**

1. Is the manuscript technically sound, and do the data support the conclusions?

Reviewer #1: Yes

2. Has the statistical analysis been performed appropriately and rigorously? 

Reviewer #1: Yes

3. Have the authors made all data underlying the findings in their manuscript fully available?

Reviewer #1: Yes

4. Is the manuscript presented in an intelligible fashion and written in standard English?

Reviewer #1: Yes

5. Review Comments to the Author

Reviewer #1: Thank you for the opportunity to review this well-written manuscript.

- Please insert line numbers.

- Methods: following IRB statement, please state that the manuscript adheres to the STROBE guidelines and provide proper citation.

- Were there any other restrictions on inclusion / exclusion?

- Spoken language

- Literacy

- Pregnancy status

- Vulnerable populations (prisoners, developmental delay/mental retardation, minority groups, etc…)

- State how the survey was developed, including whether the usability and technical functionality of the electronic questionnaire had been tested before fielding the questionnaire.

- Please provide descriptions of face validity and content validity.

- How/where was the survey announced or advertised?

- Were any incentives offered (eg, monetary, prizes, or non-monetary incentives such as an offer to provide the survey results)?

- In what timeframe were the data collected?

- To prevent biases items can be randomized or alternated. Please state if this was done.

- Where respondents were able to review and change their answers?

- Please describe the participation rate.

- Were only completed questionnaires analyzed? What threshold was used to determine completion (100%, 90%, 75%, etc.)?

- Indicate whether any methods such as weighting of items or propensity scores have been used to adjust for the non-representative sample; if so, please describe the methods.

- How was data normality determined?

- Overall, the population seemed willing to enroll in a research study despite little prior exposure or knowledge of informed consent. It would be interesting to know what the willingness to participate was amongst those persons who DID have prior exposure / knowledge of informed consent.

6. PLOS authors have the option to publish the peer review history of their article (what does this mean?). If published, this will include your full peer review and any attached files.

Reviewer #1: Yes: Andrew C. Miller

---

## [Author Response · Author response to Decision Letter 0]

26 Jun 2020

Dear Dr. Andrew Carl Miller

Academic Editor PLOS ONE

 We would like to express our full appreciation for giving us the opportunity to revise our manuscript. We believe that the reviewers made insightful suggestions that greatly helped to address critical unresolved issues to improve the manuscript. All the changes in the paper are highlighted in green in the revised version to indicate the revised portions of the manuscript. We expect that our manuscript can be now suitable for publication in PLOS ONE.

I remain at your disposal to clarify any pending point.

Yours sincerely,

On behalf of all the authors, 

Thiago D. Correa

---

## [Editor Report · Decision Letter 1]

13 Jul 2020

Intensive care unit patients’ opinion on enrollment in clinical research: a multicenter survey

PONE-D-20-10276R1

Dear Dr. Corrêa,

We’re pleased to inform you that your manuscript has been judged scientifically suitable for publication and will be formally accepted for publication once it meets all outstanding technical requirements.

Kind regards,

Chiara Lazzeri

Academic Editor

PLOS ONE